

**Study on the measurement of isoprene by Differential Optical**
**Absorption Spectroscopy**
**Song Gao[1,4], Shanshan Wang[1,2], Chuanqi Gu[1],Ruifeng Zhang[1], Yanlin Guo[1], Yuhao**
**Yan[1],Bin Zhou[1,2,3]**
[1]Shanghai Key Laboratory of Atmospheric Particle Pollution and Prevention (LAP[3]), Department
of Environmental Science and Engineering,   Fudan University, Shanghai 200438,China.
[2]Institute of Eco-Chongming (IEC), No. 20 Cuiniao Road, Shanghai 202162, China.
[3]Institute of Atmospheric Sciences,   Fudan University, Shanghai, 200433, China.
[4]Shanghai Environmental Monitoring Center, Shanghai, 200235, China.
Corresponding author: Shanshan Wang (shanshanwang@fudan.edu.cn) and Bin Zhou
(binzhou@fudan.edu.cn)
**Abstract** In this paper, the continuous on-line measurements of isoprene in the
atmosphere have been carried out by using the Differential Optical Absorption
Spectroscopy (DOAS) in the band of 202.71-227.72nm for the first time. Under the
zero optical path in the laboratory, different equivalent concentrations of isoprene
were measured by the combination of known concentration gas and series calibration
cells. The correlation between the measured concentrations and the equivalent
concentrations was 0.9996, and the slope was 1.065. The correlation coefficient
between DOAS and on-line VOCs instrument is 0.85 and the slope is 0.86 in the
comparison of 23 days field observation. It is estimated that the detection limit of
isoprene with DOAS is about 0.1ppb at an optical path of 75m, and it is verified that
isoprene could be measured in the ultraviolet absorption band using DOAS method
with high temporal resolution and low maintenance cost.
**1.  Introduction**
Isoprene, named as 2-methyl-1,3-butadiene ($C_5H_8$), is an important BVOCs
(Biological Volatile Organic Compounds) in the atmosphere. Its global emission rate
is about 500 TgCyr$^{-1}$(Sindelarova et al., 2014). Isoprene accounts for 70% of global
BVOCs emissions (Aydin et al., 2014). Land vegetation and other natural sources
contribute 90% of isoprene in the atmosphere (Zhang et al., 2016), and anthropogenic
emissions mainly come from industrial activities. Isoprene, as a typical pentadiene
hydrocarbon, has a higher activity than that of ordinary anthropogenic VOCs (Lian et





al., 2020), and its lifetime in the boundary layer is only about half an hour (Zheng et
al., 2015). Due to high volatility and reaction activity, isoprene can accelerate the
reaction between atmospheric substances, and it is easy to react with strong oxidizing
substances (OH, $NO_3$ radicals, etc.), and also affects the balance between NOx (NOx
= NO + $NO_2$) and $O_3$ in the atmosphere. Isoprene is also the precursor of secondary
organic aerosol (SOA) (Zeng et al., 2018).
Isoprene produced by plants is a byproduct of photosynthesis, its emission intensity
directly relates to the abundance of plants, leaf area index, and plant species.
Meteorological parameters, such as temperature, radiation intensity and humidity, can
also affect the emission of isoprene (Bai, 2015). In the daytime, the oxidation by OH
is the main chemical process of isoprene. Because of the existence of multiple double
bonds, the addition reaction with OH will lead to the formation of a variety of
products and the formation of $RO_2$. In the presence of NOx, $RO_2$ can be further
reacted to convert RO and $HO_2$, causing the mutual conversion of free radicals and
the accumulation of ozone, which affects the balance of $O_3$ in the atmosphere.
Meanwhile, the reaction of isoprene with $NO_3$ mainly occurs at night. Although the
reaction only accounts for 6% - 7% of the total isoprene oxidation, it is an important
way to remove $NO_3$ (Xie et al., 2013).
In recent years, with the increase of urban vegetation diversity, the emission intensity
of urban BVOCs also has a significant upward trend. The monitoring and control of
isoprene in urban ecosystem has also attracted more and more attention. Because
isoprene concentration in the atmosphere is low, and the life time is short, high
precision and accuracy methods are needed for monitoring. Currently, general
methods, including gas chromatography-mass spectrometry (GC-MS), proton transfer
reaction mass spectrometry (PTR-MS), and chemical ionization mass spectrometry
(CIMS) et al. were introduced to measure the isoprene.
GC-MS is using the high separation ability of gas chromatography to separate the
components of environmental samples, and then measuring the different compounds
with the mass spectrometry. Although GC-MS has high precision and stability, it can
distinguish most VOCs qualitatively and quantitatively. But the complex requirements



in power, temperature control and special carrier gas make it is not easy in
maintaining and operating. GC-MS measurement generally requires sampling,
preservation and pre-treatment before analysis. During this process, the sample may
change to some extent, resulting in inaccurate results.
Proton-transfer reaction mass spectrometry (PTR-MS) is the chemical ionization of
gas sample through proton transfer in drift tube. The proton source is usually $H_3O^+$.
The fixed length of the drift tube provides a fixed reaction time for the ions moving
along the drift tube, which makes the sample react with $H_3O^+$ continuously in the drift
tube to produce proton transfer, and then enter the mass spectrometer to screen
through the charge ratio. The disadvantage of PTR-MS is that it completely relies on
mass spectrometry to provide the identification of mixtures. VOCs as a class of
substances, it is possible to have the same molecular weight or the same mass of
fragment ions and parent ions. In this case, it is difficult to determine all species
present and their respective concentrations. A solution to this is to combine gas
chromatography (GC) with PTR-MS (Robert et al., 2009).
Chemical ionization mass spectrometry (CIMS) (Leibrock & Huey, 2000) retains the
qualitative ability of mass spectrometry, and coupling the traditional air sampler with
mass spectrometry technology. However, this method is not sensitive to low
concentration isoprene. In addition, the VOC composition in the atmosphere is
complex, and the unknown composition may react with benzene reagent to interfere
with the measurement results.
In addition, a portable gas chromatograph (iDirac) equipped with photo-ionization
detector to measure isoprene was proposed by Conor et al.(2020) in Cambridge
University. The instrument is an improved technology for GC-MS, which can work
independently weeks to months in the field environment. Previous studies rarely
mentioned the measurement of isoprene by spectral method. Brauer et al. (2014)
measured the infrared spectrum of isoprene by Fourier transform spectrometer, and
found that isoprene has a strong absorption near 11000nm, which provides a new
direction for the measurement of isoprene by spectral technology. So far, however,
few people have mentioned the measurement of isoprene by ultraviolet spectroscope.



In this paper, an on-line measurement method with high temporal resolution for
isoprene in the atmosphere is proposed by using the DOAS technology in the far
ultraviolet band.

## 2. Measurement method

### 2.1 Instrument introduction and spectral analysis

DOAS technology was proposed by Platt et al. (1979, 1980) in 1970s for the first time.
The principle of the instrument was detailed in other literature (Platt & Stutz, 2008),
here is the description of deep UV-DOAS. The system is mainly composed of light
source, transmitting telescope, receiving telescope, spectroscope, and computer, etc.
The transmitting and receiving telescopes are located at both ends of the measuring
optical path with a space of 75m. Since the measurement of isoprene is in deep
ultraviolet, we choose deuterium lamp (L6311-50, Hamamatsu, 35W) as light source,
the aperture of the transmitting telescope is 76mm, and the primary mirror is the UV
enhanced spherical mirror, while, the aperture of the receiving telescope is 152mm. A
spectroscope (B&W TEK Inc. BRC741E-1024) with a spectral range of 185-400 nm,
a spectral resolution of 0.75 nm FWHM (Full Width Half Maximum), and a
1024-pixel photodiode array as detector was used to record spectrum. The
measurement routine is that the light emitted by the light source is collimated by the
transmitting telescope and then sent out, after a certain distance of transmission, it is
collected by the receiving telescope and focused on the incident end of the optical
fiber. The optical fiber feeds the light into the spectroscope, which detects the light
signal and sends it to the computer for spectral analysis. The measured atmospheric
spectrum contains the absorption information of molecules in atmosphere. After
removing the Rayleigh scattering and Mie scattering, as well as the broadband
absorption of molecules by high pass filtering, the so-called differential absorption
spectrum is obtained. The concentration of the corresponding atmospheric
components can be retrieved by fitting differential absorption spectrum with the
differential absorption cross section of the measured molecules.
Isoprene has strong absorptions between 200.0-225.0nm, among which there are





relatively obvious absorption peaks (Martins et al., 2009) near 210.0nm, 216.0nm and
222.1nm, as shown in Figure 1a. After a 5th order polynomial fitting filtering, the
differential absorption spectrum (1ppb*km) of isoprene is shown in Figure 1b.
According to its differential absorption characteristics, the fitting band of isoprene is
202.71-227.72nm. Within this band, there are also absorptions of $NH_3$ (Chen et al.,
1999), $SO_2$ (Wu et al., 2000), NO, $NO_2$ (Mérienne et al., 1995), $C_6H_6$ (Dawes et al.,
2017), $C_7H_8$ (Serralheiro et al., 2015), etc. The absorption of NO used here was
measured in laboratory with known concentration gas by using the same instrument.
Therefore, the absorption of these components is also considered in the process of
spectral retrieving. Figure 1c is an example of the spectrum fitting, the black line is
the actual atmospheric spectrum (2018-07-08 12: 47), while the red line is the fitting
spectrum (0.79ppb isoprene, 2.83ppb $NH_3$, 1.85ppb $SO_2$, 1.42ppb NO, 4.94ppb $NO_2$,
0.01ppb $C_6H_6$, 2.20ppb $C_7H_8$), and figure 1d is the fitting residual (standard deviation
is 4.76E-4).
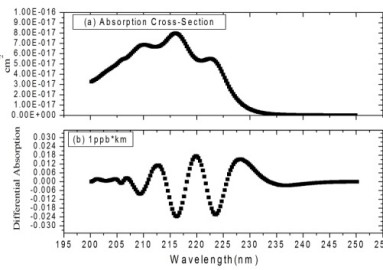 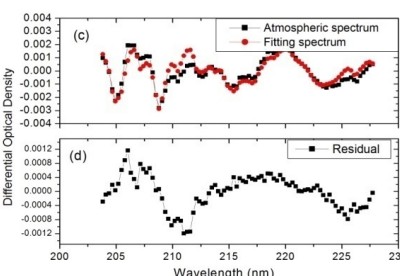
Figure1. The absorption Cross-section of isoprene(a), the differential absorption
spectrum of isoprene in 1ppb*km(b), the example of the spectrum fitting, the black
line is the actual atmospheric spectrum (2018-07-08 12: 47), and the red line is the
fitting spectrum (0.79ppb isoprene, 2.83ppb $NH_3$, 1.85ppb $SO_2$, 1.42ppb NO, 4.94ppb
$NO_2$, 0.01ppb $C_6H_6$, 2.20ppb $C_7H_8$) (c), the fitting residual (d)
**2.2Calibration experiment**
In order to verify the accuracy of measurement results, isoprene gas with known
concentration is used to calibrate the instrument in the laboratory. The method is to
close the emitting telescope and receiving telescope (close to zero optical path) in the
laboratory, and then a series absorption cell was placed between the telescopes.




10ppm isoprene gas was injected into the cells at a constant flow rate of 100ml/min,
and then the corresponding concentration under different cell combinations was
measured, as shown in Figure 2.

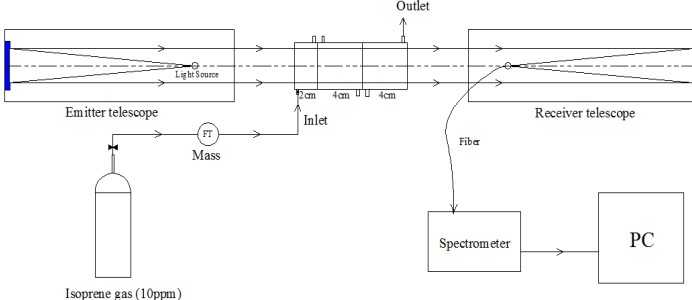


151           Figure2. The scheme of the calibration system

The absorption cell group is composed of one 2cm and two 4cm long cells in series.
When using different combination of cells, different equivalent concentrations
($C_E$)(equivalent to the average concentration in the 100m optical path) can be
obtained. The specific combination and corresponding equivalent concentrations, as
well as the actual measurement concentrations ($C_M$) are shown in table 1.

157        Table 1: the calibration results in different gas cells combination

| Length of cells | $C_E$(ppb) | $C_M$(ppb) |
|---|---|---|
| empty | 0 | 0.01 |
| 2cm | 2.00 | 1.88 |
| 4cm | 4.00 | 3.61 |
| 2cm+4cm | 6.00 | 5.40 |
| 4cm+4cm | 8.00 | 7.44 |
| 2cm+4cm+4cm | 10.00 | 9.42 |

Fig. 3 is the linear fitting of calibration results. The ordinate in the figure is the
equivalent concentration, and the abscissa is the measured concentration. For six
measuring points including the zero point, the linear fitting correlation coefficient R is
0.9996. The relationship between the equivalent concentration and the measured
concentration is shown in the following equation (1). For the future measurement
results of the actual atmosphere, equation (1) will be used to calibrate the measured
data.

165                $C_E = 0.06\text{ppb} + 1.066 * C_M$ (1)


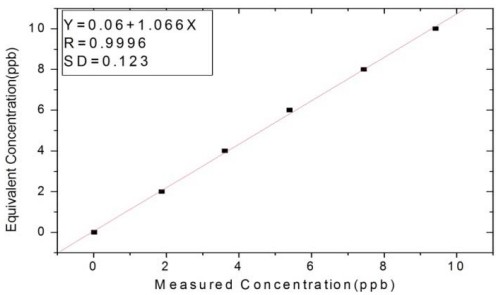


Figure3. The linear fitting of calibration results, the ordinate is the equivalent
concentration and the abscissa is the measured concentration
**3. Field comparison experiment and discussion**
**3.1Comparison with on-line VOCs results**
In order to further verify the reliability of DOAS method in actual atmospheric
measurement, in July 2018, the field measurement results of the DOAS were
compared with the on-line VOCs (TH-300B on-line VOCs monitoring system)
analyzer (Zhu et al., 2020), which is based on the GC-MS technology. DOAS
instrument is installed on the 7th floor of the Environmental Science Building
(31.344 ° N, 121.518 ° E) in Jiangwan campus of Fudan University, as shown in
Figure 4. The optical path is about 25m above the ground. The transmitting telescope
is at the west part of the building (A in Figure 4), while the receiving telescope is at
the east part (B in Figure 4). The distance between the telescopes is 75m. The on-line
VOCs instrument is located in Xinjiangwan City monitoring station of Shanghai
Environmental Monitoring Center (C in Figure 4). The straight-line distance is about
0.5km to the south of DOAS instrument. The coverage rate of plants around the
observation sites is high, mainly including pine, camphor, etc., and a large number of
lawns are also distributed. Meteorological parameters were recorded by the automatic
weather station(CAMS620-HM, Huatron Technology Co. Ltd) co-located with DOAS
instrument.



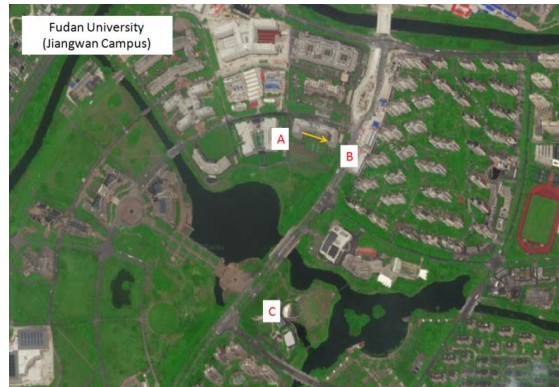


Figure4. Field measurement sites of DOAS and on-line VOCs, A is the transmitting
telescope, B is the receiving telescope, and C is the on-line VOCs, the yellow arrow is

190                    light path of DOAS. This map is sourced from © Baidu

The comparison experiment was carried out from July 1$^{st}$ to 23$^{rd}$, 2018. The temporal
resolution of DOAS was 1 min, while that of on-line VOCs was 1 h. In order to make
a good comparison, DOAS data were averaged hourly. Figure 5 shows the time series
of the data. It can be seen that the measurement results of the two instruments are in
good agreement. The average values of DOAS and on-line VOCs were 0.325ppb and
0.217ppb, and the standard deviation (SD) was 0.254ppb (N=551) and 0.257ppb
(N=466), respectively. The average value of DOAS results is higher than on-line
VOCs mainly because, at night, DOAS can still detect a certain concentration in most
cases, most of which are between 0.02-0.1ppb, while most of on-line VOCs data are
between 0-0.05ppb. Since the observation is in summer, there is also a very high
temperature at night during the observation period, i.e. 27.1℃ (19:00-06:00 next
morning). In addition, the release of isoprene produced by the leaves of plants in the
daytime is delayed to some extent, resulting in a certain concentration of isoprene
remaining at night, so that we think the data of DOAS is more reasonable.

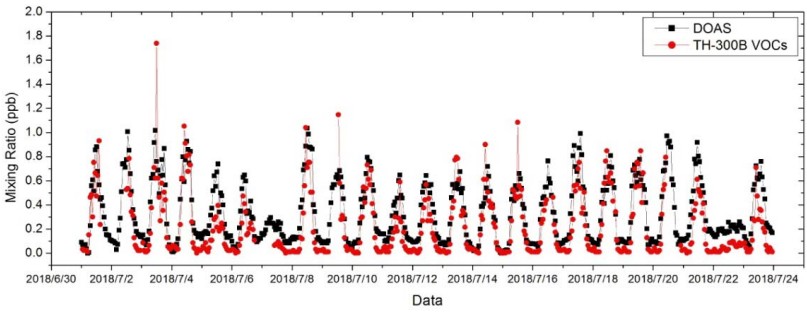


Figure5. Time series data of DOAS and on-line VOCs measured isoprene during the

comparison measurement

Due to the missing of some data of on-line VOCs during the comparison period,
totally 466 sets of hourly data were used to analyze the correlation between these two
instruments. As shown in Figure 6, the correlation coefficient is 0.85 and the slope is
0.86. The main reason for the difference of DOAS and on-line VOCs results is that
the sampling and measurement heights of the two instruments are different. The light
path of DOAS is about 25m above the ground, while the sampling height of on-line
VOCs instrument is about 10m. Isoprene will rise up and diffuse after emission from
plants, so higher measurement points will catch higher concentration of isoprene. In
addition, the sampling of on-line VOCs is through the sampling tube, and isoprene
will be more or less lost during the sampling process. Those two reasons will
eventually lead to DOAS measurement results higher than online VOCs instruments,
especially when the isoprene concentration is very low at night, the difference is more
obvious.
It can also be seen from Figure 6 that when the isoprene concentration is higher than
0.5ppb, the measurement results of the two instruments show large scattering. The
main reason is that the spatial distance between the two instruments is about 500m,
considering the inhomogeneity spatial distribution of isoprene, this spatial difference
will lead to different data results between two instruments. Meanwhile, there are
various vegetations between the instruments, when the wind direction changes, the
emission of this part of vegetation will also cause the difference between the results of





the instruments. The different measurement principles, especially the difference of
sampling time can also cause the scattering of the results of two instruments. On-line
VOCs only has about 50% of the time (1h) to be used to sampling, while the rest of
the time is used for analysis. But DOAS is almost continuous measurement with just a
little part of time to be used for analysis (about 1s per minute), this difference will
affect the consistency of results. But in general, DOAS and on-line VOCs analyzers
show a good agreement in the comparison of mean and correlation of measured data.

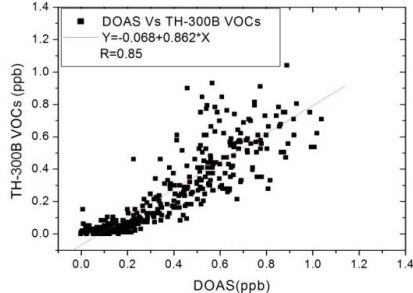


Figure6. The correlation between DOAS and on-line VOCs instruments during the

237                 field measurement

**3.2 Detection limit evaluation**
The detection limit of DOAS mainly depends on the signal-to-noise ratio of the
spectrum. Under the condition of zero light path in the laboratory, the zero noise
(standard deviation of the results) of isoprene is 0.007ppb, the detection limit can be
definite as two times of zero noise, so that the detection limit of the system is
0.014ppb (HJ 654-2013). However, in the real atmospheric measurement, it is
difficult to determine the actual detection limit due to the varied environment and the
interference of other gases. The detection limit of DOAS in real atmosphere is mainly
determined by the residual of spectral fitting. The residual mainly comes from the
absorption of interfering substances, the change of lamp spectral intensity and
structure, the spectral shift caused by the change of ambient temperature of the
spectrometer, and the noise of the detector. In order to reduce the influence of these
factors on the measurement, in the process of spectra fitting, the absorption of
interfering substances and the spectral structure of lamp are necessary to be



considered together with the isoprene. At the same time, it is also necessary to
calibrate the spectral drift. However, there are still some residual remain after the
spectral fitting.
In the fitting band of isoprene, the absorption of NO, benzene and toluene are the
main interference factors. The reason for the influence of NO is that there are three
obvious absorption peaks of NO in the fitting band. After high pass filtering, there is
component in the differential absorption cross section of NO similar to the variation
frequency of isoprene's differential absorption spectrum. After the analysis of the
measurement results, the impact of NO on isoprene is about 0.3% of its concentration.
But the effect of NO is mainly in the morning and evening rush hour. The influence of
benzene and toluene is mainly due to their strong absorptions in the fitting band of the
spectrum. Their presence will lead to a significant reduction in the spectral intensity in
the band, resulting in a reduction in the signal-to-noise ratio of the spectrum. During
the comparison experiment, high concentration of benzene or toluene occasionally
occurs, resulting in a large fitting residual. Other aromatics, such as xylene and
styrene, also absorb strongly in the fitting band, but because of their lower
concentration in the natural atmosphere, their impacts on isoprene are significantly
smaller than that of benzene and toluene. Although $NH_3$, $SO_2$ and $NO_2$ have
absorption in the fitting band, their differential absorption variation frequency is
significantly higher than that of isoprene, and only overlaps in parts of fitting band, so
that they have little influence on the isoprene measurement. Fig. 7a is the absorption
cross section of benzene, toluene and isoprene, while Fig. 7b is the differential
absorption spectra (1ppb*km) of NO, $SO_2$, $NO_2$, $NH_3$ and isoprene.

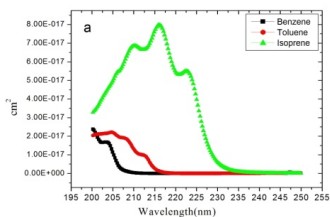 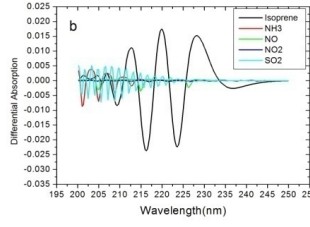


Figure7.The absorption cross section of benzene, toluene and isoprene (a), the





differential absorption spectra ((1ppb*km) of NO, $SO_2$, $NO_2$, $NH_3$ and isoprene (b)
Whether it is benzene, toluene, or NO, $SO_2$, $NO_2$ and $NH_3$, they all exist together with
isoprene in the atmosphere. Therefore, their influences on isoprene measurement are
common. In order to ensure the quality of results, the data with a residual of more
than 0.0005 are filtered out. In a total of 33120 sets of data during 23 days observation,
1137 sets are filtered out, and the valid rate of data is 96.6%. The average residual of
all valid data is 0.000234. In order to evaluate the detection limit of DOAS in the real
atmospheric measurement, we made a statistic on 16387 sets of data with the
concentration of isoprene lower than 0.1ppb (assuming that the isoprene in the
atmosphere is close to zero at this time), and the standard deviation is 0.0499ppb, so
the detection limit of DOAS instrument in the field measurement is no more than
0.1ppb (twice the standard deviation).
**4. Conclusion**
This paper introduces, for the first time, the continuous on-line measurement of
isoprene in the atmosphere by means of DOAS in the band of 202.71-227.72nm.
Although the current measurements of isoprene are mainly GC-MS, PTR-MS and
CIMS methods, The DOAS method has the characteristics of high time resolution,
rapid temporal response and simple operation. It is especially suitable for long-term
online measurement in the field or forest where the traffic is inconvenient, and the
low cost of instrument is also conducive to build monitoring network.
Under the condition of zero optical path in the laboratory, several equivalent
concentrations were measured by using a series absorption cell and known
concentration of isoprene gas. The correlation coefficient between the measured
concentration and the equivalent concentration was 0.9996, and the slope was 1.065,
indicating that the instrument has good linearity and accuracy. After 23 days of field
comparison, there is a good correlation between the results of DOAS and on-line
VOCs instrument, with a correlation coefficient of 0.85 and a slope of 0.86.
Considering the different measurement principles, the different measurement
environment and the space distance between them, the comparison results shows a





good agreement between the two instruments.
In order to evaluate the detection limit of DOAS instrument under the actual
atmospheric measurement, the paper proposes to calculate the standard deviation of
all the data when the measured concentration of isoprene in the ambient air is close to
zero (< 0.1ppb, n = 16387). It is estimated that the detection limit of the DOAS is no
more than 0.1ppb under a measurement light path of 75m. Therefore, the DOAS is
suitable for long-term monitoring in cities or areas with large vegetation coverage.

**Data availability**. Data are published as https:// DOI: 10.17632/489mvgbsxg.3
**Author contribution.** The study was designed by SG and BZ. Experiments were
performed by YG, RZ and YY. Data processing and analysis were done by BZ and
CG. The paper was written by BZ , SW and SG.
**Competing interests.** The authors declare that they have no conflict of interest.
**Acknowledgements.** This research has been supported by the National Key Research
and Development Program of China (grant No. 2016YFC0200401and
2017YFC0210002), the National Natural Science Foundation of China (grant No.
21777026, 41775113 and 21976031).

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
