# Peer review of "Study on the measurement of isoprene by Differential Optical"

_Atmospheric Measurement Techniques, 2020_

## Referee Comment (RC1) · Anonymous Referee #1 · 26 Nov 2020

The manuscript of Sao Gong and co-authors is a very interesting work on new kind of isoprene measurements using active DOAS (Differential Optical Absorption Spectroscopy) in the deep UV spectral range between 200 and 230nm. It is to my knowledge the first manuscript presenting so clearly the possibilities for quantitative isoprene measurements with this technique and fit well in the scope of AMT. The manuscript contain all basic information on instrumentation, characterization, data analysis and a field comparison experiment. Some of the information are incomplete and should be corrected. I recommend the publication of this manuscript after correction of the following points:

Major points:

1. Section 2.1 after line 115: It is not clear how the Reference I0 spectrum of the light

source is recorded and how it is considered in the spectral fit. The shape and spectral properties should be considered. How I0 measurements are made in measurements of section 3.1?

2. l. 128: NO absorption cross section was recorded with a reference gas. At which concentration and slant column (cell length). Is the NO signal representative for the atmospheric concentration over 75m? NO absorption contain narrow absorption bands which go into saturation. If the NO concentration is not representative these saturation effects should be considered.

3. Please explain in more detail the spectra preparation before the DOAS fit.

4. l. 133 and following: Provide DOAS fit errors for the derived concentrations. You may use the method of Stutz, J. and Platt U., Numerical analysis and estimation of the statistical error of differential optical absorption spectroscopy measurements with least-squares methods, Appl. Opt., 35 (30), 6041-53, 1996. Include errors of the measurements also in section 3 and 4.

5. Figure 1: Include all fitted reference gases.

6. l. 165 show a 6.6% underestimation. Please include an error estimation. Is this within the error? What are possible reasons? Could it be due to spectrometer stray light? Did you check the spectrometer stray light below 230nm?

7. l. 239: Your DOAS fit like in Fig. 1 is not dominated by noise but by a remaining spectral structure. This is dominating here your error. Apply a proper DOAS error calculation (see comment above). You can not simply translate the zero noise to a measurement noise over 75m. In the later case you have also interference's from other gases and imperfect spectral data, and likely also changes of our light source. The zero noise estimation is a lower limit and true error will be higher.

Minor points:

l. 101: include reference "(see Figure 2)"

l. 105: "the aperture of the transmitting telescope is 76mm, and the primary mirror is the UV enhanced spherical mirror...". - What is the focal length. Rephrase to make the sentence clear e.g.: The aperture of the transmitting telescope is 76mm, with a UV enhanced spherical mirror with a focal length of XXXmm. The aperture of the receiving telescope is 152mm, with a XXX mirror with a focal length of XXXmm.

l. 116: Specify the high pass filter.

l. 147: include the accuracy of the isoprene concentration in the bottle.

l. 198 – 199: The isoprene observed by DOAS at night is most of the time significantly above zero and does not reach zero later in the night. Could this not be a systematic offset e.g. due to missing I0 reference spectra?

l. 212: Here the argument for the difference is the sampling height. Before it was the night measurement. Put both arguments together in a merged explanation.

l. 214: "Isoprene will rise up and diffuse after emission from plants, so higher measurement points will catch higher concentration of isoprene." → This is completely wrong. Concentration can not accumulate and will always be same or lower with larger distance from the source.

l. 216: Specify expected losses in sampling tubes.

l. 217: Include estimation of (systematic) DOAS errors and if this can explain the difference.

l. 235: include measurement errors as example.

l. 251: It is not explained how the spectral structure of the lamp is corrected.

l. 254: "remain after the spectral fitting" , Include why they include → due to. . .. imperfect reference spectra

l. 260: How do you derive the influence of NO on isoprene measurements?

l. 274: Define the high pass filter for the differential absorption spectra. Is it the same like for the spectral analysis?

l. 275: Figure 7a), why these spectra are not shown as differential spectra?

Language:

l. 19 – 21, rephrase: The correlation coefficient between DOAS and on-line VOCs instrument observed from 23 days field observation is 0.85 with a slope of 0.86.

l. 43 – 45, rephrase: Due to the existence of multiple double bonds, the additional reaction with OH will lead to the formation of a variety of products and the formation of RO2.

l. 58: measure the isoprene → meausre isoprene

l. 61: spectrometry → spectrometer

l. 63: is not easy → difficult

l. 91: direction → possibility

l. 103: space → separation or distance

l. 104: source, → source.

l. 109: record spectrum → record the spectrum

l. 115: in atmosphere → in the atmosphere

l. 119: fitting differential → fitting the differential

l. 158: Fig. 3 is the linear fitting of the calibration→ Fig. 3 shows the linear fit of the calibration

l. 162: For the future → For further

l. 171 & l. 182: of DOAS → of the DOAS

l. 185: with DOAS → with the DOAS

l. 189: is lightpath → is the lightpath

l. 196: 0.217ppb, → 0.217ppb respectively,

l. 197: that on-line → that of the on-line

l. 252: isoprene. → isoprene absorption spectrum.

---

## Referee Comment (RC2) · Anonymous Referee #2 · 30 Nov 2020

The manuscript entitled "Study on the measurement of isoprene by Differential Optical Absorption Spectroscopy" by Song et al. reports the application of DOAS technique on isoprene measurement. This study details the setup, laboratory experiments, and field applications of the DOAS. Intercomparisons of isoprene concentrations measured by the DOAS and a commercial GC-MS shows a good consistency. The content and novelty of the manuscript align well with the requirements of AMT. However, English could be improved throughout the paper as it is not easy to follow. Overall, I recommend the manuscript for publication if the authors can address the following comments.

General Comments: 1. In the calibration experiments described in Sect. 2.2, the actual concentrations (CM) must be measured more than once with parallel experiments for different cell lengths, and measuring error should be added in Table 1 and Figure 3.

The equation (1) should be recalculated accordingly. In addition, it seems that the difference between CE and CM increased as the increase of cell length in Table 1. Could authors provide an explanation about the phenomenon?

2. Some important information is missing in the comparison experiments. Firstly, it is not given how reference spectrum was recorded during field applications. As reference spectrum plays a role in spectral fitting, uncertainty caused by reference spectrum should be discussed. Secondly, calibration methods as well as calibration frequency of the on-line VOCs (TH-300B) are not provided. Compared with the DOAS measurements, the on-line VOCs measurements seems to have a 0.1 ppb offset during the period from 07/21 to 07/24. Could the offset be caused by the calibration of on-line VOCs? Thirdly, providing wind parameters (measured by weather station) and benzene and toluene concentrations (measured by on-line VOCs) when the comparison is inconsistent will be more persuasive, as authors speculated that wind directions and benzene and toluene concentrations would influence the comparison consistency.

3. As authors introduced in Sect. 1, PTR-MS and CIMS can also be used to measure isoprene concentrations. The manuscript would benefit from a critical comparison of the best available performance of these four methods (i.e., DOAS, GC-MS, PTR-MS, and CIMS) together given in a table. Characteristics in the comparison could be time resolution, accuracy, precision, appropriate platforms, etc. Such a comparison would be useful to the readership and meaningful to the community.

Specific Comments: Line 15: "202.71-227.72nm" → "202.71-227.72 nm". Blank space should be inserted between number and unit. Such irregular expressions were used frequently elsewhere in the manuscript and should be revised.

Line 26: The "B" in BVOCs is usually the abbreviation of "Biogenic" instead of "Biological".

Line 42: "In the daytime, the oxidation by OH is the main chemical process of isoprene." The sentence should specify "whose oxidation" and "what kind of chemical process" to

avoid ambiguous meaning.

Line 44-47: These contents are given here without references.

Line 52: "BVOCs also has [. . .]" → "BVOCs has [. . .]"

Line 59: "GC-MS is using the high separation ability of gas chromatography to separate the [. . .]". Simple Present Tense should be used here.

Line 61-64: "Although GC-MS [. . .] But the complex [. . .]" These sentences should be rephased.

Line 63ff: A comma should go before the conjunction "and" in a list of three or more items. "[. . .] in power, temperature control and special carrier gas [. . .]" → "power, temperature control, and special carrier gas". "[. . .] requires sampling, preservation and pre-treatment [. . .]" → "requires sampling, preservation, and pre-treatment".

Line 67, 77, and 78: The meaning of the abbreviations (i.e., PTR-MS, GC, and CIMS) has already been given in the previous paragraph and so it need not be defined again here.

Line 69-71: The sentences should be rephased.

Line 104: "[. . .] as light source, the aperture [. . .]" → "as light source. The aperture"

Line 106: "while, the aperture of the" → "while the aperture of the"

Line 109: "1024-pixel photodiode array as detector was used to record spectrum" → "1024-pixel photodiode array was used as detector to record spectrum"

Line 158: "Fig. 3" → "Figure 3"

Line 215: "[. . .] so higher measurement points will catch higher concentration of isoprene. [. . .]" Reference or detailed explanations should be given here.

Line 217: "[. . .] will be more or less lost during the sampling process." Sampling loss of on-line VOCs is an important parameter which should be quantized here by performing

experiments or referring to a similar research.

Line 241-243: The authors should provide an explanation or references on the method that they used to calculate detection limit.

Line 247-249: As the stability of light source and spectrometer will influence the fitting residual and instrumental performance, sensitivity experiments of temperature (or other relative parameters) for light source and spectrometer should be conducted.

Line 293: "CIMS methods, The" → "CIMS methods, the"

Line 304-306: The sentences should be rephased.

Line 308: "the paper proposes [. . .]" → "this study proposes [. . .]"

---

## Author Comment (AC1) · 19 Jan 2021

**Response to reviewers' comments #1**

We thank the reviewers for the constructive comments and suggestions, which are very positive to improve scientific content of the manuscript. We have revised the manuscript appropriately and addressed all the reviewers' comments point-by-point for consideration as below. The remarks from the reviewers are shown in black, and our responses are shown in blue color. All the page and line numbers mentioned following are refer to the revised manuscript without change tracked.

Anonymous Referee #1

The manuscript of Sao Gong and co-authors is a very interesting work on new kind of isoprene measurements using active DOAS (Differential Optical Absorption Spectroscopy) in the deep UV spectral range between 200 and 230nm. It is to my knowledge the first manuscript presenting so clearly the possibilities for quantitative isoprene measurements with this technique and fit well in the scope of AMT. The manuscript contain all basic information on instrumentation, characterization, data analysis and a field comparison experiment. Some of the information are incomplete and should be corrected.

I recommend the publication of this manuscript after correction of the following points:

Major points:

1. Section 2.1 after line 115: It is not clear how the Reference I0 spectrum of the light source is recorded and how it is considered in the spectral fit. The shape and spectral properties should be considered. How I0 measurements are made in measurements of section 3.1?

R: Thank you for the comments. We have supplemented the related description about how the reference I0 spectrum is recorded in laboratory experiments and filed measurements, respectively. Please refer to Line 104-107 in the revised manuscript. The consideration about I0 spectrum interference was stated in Line 246-248 in the manuscript.

2. l. 128: NO absorption cross section was recorded with a reference gas. At which concentration and slant column (cell length). Is the NO signal representative for the atmospheric concentration over 75 m? NO absorption contain narrow absorption bands which go into saturation. If the NO concentration is not representative these saturation effects should be considered.

R: The guaranteed NO gas with concentration of 3080 ppm was used to record the reference spectrum using 0.02 m cell length, which is equivalent to about 820 ppb with 75 m light path during the field measurement. Although this NO signal is not representative for the atmospheric concentration, we have performed the measurements for different NO concentrations in order to testify this NO reference can be employed for the spectral fitting of the field observation.

Considering the ambient NO levels, a series of spectra containing equivalent NO concentrations of 0, 40, 80, 160, 200 ppb under 75 m light path have been measured,

respectively. Each concentration points have been measured repeatedly multi times, as summarized in Table R1.

Table R1. The calibration results of NO in different gas cells combination

| Length of cells | $C_E$ (ppb) | $C_M$ (ppb) |
|---|---|---|
| empty | 0 | 0.91±1.04 |
| 2 cm | 40.00 | 40.94±0.78 |
| 4 cm | 80.00 | 81.02±0.70 |
| 2 cm + 4 cm | 120.00 | 120.85±0.76 |
| 4 cm + 4 cm | 160.00 | 160.71±0.99 |
| 2 cm + 4 cm + 4 cm | 200.00 | 199.16±0.94 |

Figure R1 shows the differential optical density for the equivalent NO concentration series and the correlation between equivalent and measured concentration of NO. It can be seen that the measured NO for different equivalent concentrations would not be interfered by using the reference recorded at high concentration. And the measured NO concentrations were highly consistent with the equivalent concentration showing a slope of 1.01 and correlation coefficient R2 of 1.

[Figure]

*Figure R1. Differential optical density for the equivalent NO concentration series and the correlation between equivalent and measured concentration of NO.*

Therefore, we consider that the NO reference is suitable used for the spectral analysis of the atmospheric measured spectrum containing less NO concentration.

3. Please explain in more detail the spectra preparation before the DOAS fit.
R: Before the DOAS fit, the measured spectra with low light intensity and high integration time were excluded from the spectral fitting, which are mainly due to the unfavorable weather condition influencing the measurements. The spectral were also corrected for offset before introducing fitting. Please refer to Line 171-172 in the revised manuscript. Regarding to the absorption cross sections of NH3, SO2, NO2,

C6H6 and C7H8, these high-resolution references were convoluted with a Gaussian-shaped instrument function of 0.75 nm half-width to obtain the absorption cross section matching the spectrometer resolution. Please refer to Line 114-115 in the revised manuscript.

4. l. 133 and following: Provide DOAS fit errors for the derived concentrations. You may use the method of Stutz, J. and Platt U., Numerical analysis and estimation of the statistical error of differential optical absorption spectroscopy measurements with least-squares methods, Appl. Opt., 35 (30), 6041-53, 1996. Include errors of the measurements also in section 3 and 4.

R: Thank you for the suggestions. We have carefully reviewed this literature and followed the method to estimate the DOAS fit errors of this study (Stutz, J. and Platt U., 1996). Overall, the measurement errors of isoprene were estimated lower than 20%. The errors have been indicated in Line 126-127 in the revised manuscript. Besides, the errors of the measurements were also included in section 3 and 4, and summarized in Line 229.

5. Figure 1: Include all fitted reference gases.

R: Figure 1(a) and (b) shows the absorption cross section and differential absorption spectrum of isoprene in 1 ppb*km. Other interference absorption gases were showed in Figure 7. So we speculate the reviewer suggested to include all the references in the spectral fitting example (Fig. 1(c) and (d)). We have re-plotted it, as shown in Figure 2 in the revised manuscript.

6. l. 165 show a 6.6% underestimation. Please include an error estimation. Is this within the error? What are possible reasons? Could it be due to spectrometer stray light? Did you check the spectrometer stray light below 230nm?

R: In the revised manuscript, the measuring errors have been included in the Table 2 and linear regression analysis. The 6.6% underestimation was determined from the linear regression equation. We think this underestimation may be due to the possible tiny bias in the length of cell or the uncertainty of the standard gas sample, rather than the spectrometer stray light. The spectrometer stray light is not exceeding 0.8% around 200 nm band (as mentioned in the spectrometer instruction). We would not attribute this underestimation to the impacts of stray light of spectrometer.

7. l. 239: Your DOAS fit like in Fig. 1 is not dominated by noise but by a remaining spectral structure. This is dominating here your error. Apply a proper DOAS error calculation (see comment above). You can not simply translate the zero noise to measurement noise over 75m. In the later case you have also interference's from other gases and imperfect spectral data, and likely also changes of our light source. The zero noise estimation is a lower limit and true error will be higher.

R: As responses above, we have followed the method to estimate the DOAS fit errors of this study (Stutz, J. and Platt U., 1996).

Minor points:

l. 101: include reference "(see Figure 2)"

R: We have added it in Line 87 of the revised manuscript.

l. 105: "the aperture of the transmitting telescope is 76mm, and the primary mirror is the UV enhanced spherical mirror...". - What is the focal length. Rephrase to make the sentence clear e.g.: The aperture of the transmitting telescope is 76mm, with a UV enhanced spherical mirror with a focal length of XXXmm. The aperture of the receiving telescope is 152mm, with a XXX mirror with a focal length of XXXmm.

R: Thank you for the comments. These sentences have been rephrased as "The aperture of the transmitting telescope is 76 mm, with a and the primary mirror is the UV enhanced spherical mirror with a focal length of 304 mm. The aperture of the receiving telescope is 152 mm with a UV enhanced spherical mirror with a focal length of 608 mm". Please refer to Line 90-92 in the revised manuscript.

l. 116: Specify the high pass filter.

R: In this study, high pass filter is to perform a high pass binomial on the spectrum using the iterations of 500 twice. The operation will first do a low pass filter and will then divide the spectrum by the result of this low pass filter operation. The binomial filter uses the smallest binomial mask possible. This mask does the same as an averaging operation over three contiguous channels. Afterwards, the broadband structures can be eliminated effectively. Please refer to Line 101-102 in the revised manuscript.

l. 147: include the accuracy of the isoprene concentration in the bottle.

R: According to the certificate of reference material of this gas, the uncertainty of isoprene gas with 10 ppm (guaranteed values of standard samples) is 2% (confidence interval of 0.95). Please refer to the revised Figure 3.

l. 198 – 199: The isoprene observed by DOAS at night is most of the time significantly above zero and does not reach zero later in the night. Could this not be a systematic offset e.g. due to missing I0 reference spectra?

R: During the field measurement, the measured spectrum collected at 00:00 LT on July 1, 2018 was used as the reference spectrum. If the reference spectrum is pure enough without any absorption of isoprene, the DOAS retrieved data would be accurate even though it is most of the time significantly above zero at night and does not reach zero later in the night. Given that the reference spectrum was contaminated by the rare isoprene absorption, the DOAS retrieved data would be lower than the real value. The observed isoprene should be even higher in the night. On the other hand, the VOCs analyzer data series showed the isoprene concentrations were close to zero at night, which may be related to the daily calibration procedure operated at 00:00-01:00. So we would like to attribute this systematic offset to VOCs analyzer, rather than the offset from I0 spectrum.

l. 212: Here the argument for the difference is the sampling height. Before it was the night measurement. Put both arguments together in a merged explanation.

R: Thank you for the suggestions. We have re-structured this part in order to achieve a better merged explanation. Please refer to Line 186-213 in the revised manuscript.

l. 214: "Isoprene will rise up and diffuse after emission from plants, so higher measurement points will catch higher concentration of isoprene." → This is completely wrong. Concentration can not accumulate and will always be same or lower with larger distance from the source.

R: Thank you for pointing out this wrong argument. We have removed this explanation from the text and try to explain the possible reasons causing the discrepancies between these two instrumental observed isoprene values. Please refer to Line 186-213 in the revised manuscript.

l. 216: Specify expected losses in sampling tubes.

R: The TH-300B on-line VOC instrument uses detection technology that includes ultralow-temperature preconcentration combined with gas chromatography and mass spectrometry (GC/MS). For a complete measuring cycle, there five steps include sample collection, freeze-trapping, thermal desorption, GC-FID/MS analysis, heating and anti-blowing purification. It takes about 1h for one complete detection cycle. It's extremely difficult to evaluate the sampling loss individually from the complete detection cycle, especially for experiments. As a commercial scientific instrument, the relative error of the targets was quantized not exceeding 30% among all the detected VOCs species for the whole procedure, which is determined by the difference between the measured and guaranteed values of standard samples. As to isoprene, the $R^2$ reached 0.999 (Hui et al., 2019). It suggests that the detection accuracy is very high, and the total error could be contributed considerably by sampling process (EPA, 2019). We have addressed the related discussion in Line 191-192 in the revised manuscript.

l. 217: Include estimation of (systematic) DOAS errors and if this can explain the difference.

R: Regarding to the differences in the observed isoprene by these two instruments, especially for night, we have discussed from several aspects, e.g. instrumental principle, sampled air, impacts of meteorological conditions, etc. Please refer to Line 186-213 in the revised manuscript.

l. 235: include measurement errors as example.

R: There is Figure 6 in Line 235. We are not sure if the reviewer suggested to include the measurement errors in this Figure. We have indicated the errors, as shown in Figure 6(b) in the revised manuscript.

l. 251: It is not explained how the spectral structure of the lamp is corrected.

R: We have not corrected the spectral structure of the lamp, however, included the lamp spectrum as the absorption of interfering substances in the spectral fitting. In this way, the impacts due to spectral structure of the lamp can be reduced significantly and the residual is consequently much lower. Therefore, we have stated that "the absorption of interfering substances and the spectral structure of lamp are necessary to be considered together with the isoprene absorption spectrum". Please refer to Line 226-227 in the revised manuscript.

l. 254: "remain after the spectral fitting", Include why they include → due to….

imperfect reference spectra

R: Thank you for the suggestions. We have included the possible reason causing the residual after spectral fitting. Please refer to Line 228-229 in the revised manuscript.

l. 260: How do you derive the influence of NO on isoprene measurements?

R: Given the absorption of NO can influence the isoprene absorption, it can be found that the occurrences of NO peak value are sometimes consistent with isoprene, as the example of a short period shown in Figure R2 (spectral temporal resolution). By investigating the spectral fitting, there were no obvious absorption of isoprene. If the isoprene concentration was subtracted by 0.3% NO concentration, some weird isoprene peaks will be disappeared. Therefore, we have inferred that the impact of NO on isoprene could be about 0.3% of its concentration.

[Figure]

*Figure R2. Time series of NO and isoprene concentration.*

l. 274: Define the high pass filter for the differential absorption spectra. Is it the same like for the spectral analysis?

R: Yes, the high pass filter for the differential absorption spectra is applied same as this of the spectral fitting. We have defined it in Line 245-246 in the revised manuscript.

l. 275: Figure 7a), why these spectra are not shown as differential spectra?

R: In order to display the absorption of benzene and toluene in deep UV wavelength, we show the absorption cross sections of benzene and toluene together with isoprene. It can be found that the measured light intensity below 215 nm will be reduced significantly if the benzene and toluene concentration are high, which will further reduce the signal to noise ratio of the measured spectra and influence the spectral fitting performance. The aim of Fig. 7(b) is demonstrating the variation frequencies of differential absorption of NH3, SO2 and NO2 are much higher than that of isoprene. Therefore, the interferences of NH3, SO2 and NO2 absorption on isoprene can be weakened in the fitting process of differential spectra. Considering as mentioned above, we decided to display these two patterns of the absorption spectra in Figure 7 (a) and (b).

Language:

l. 19 – 21, rephrase: The correlation coefficient between DOAS and on-line VOCs instrument observed from 23 days field observation is 0.85 with a slope of 0.86.

R: Thank you for the comments. We have followed the suggestion to rephrased as "The correlation coefficient between DOAS and on-line VOCs instrument observed from 23 days field observation is 0.85 with a slope of 0.86". Please refer to Line 18-19 in the revised manuscript.

l. 43 – 45, rephrase: Due to the existence of multiple double bonds, the additional reaction with OH will lead to the formation of a variety of products and the formation of RO2.
R: Thank you for the comments. We have followed the suggestion to rephrased as "Due to the existence of multiple double bonds, the additional reaction with OH will lead to the formation of a variety of products and the formation of RO2". Please refer to the Line 36-38 in the revised manuscript.

l. 58: measure the isoprene → meausre isoprene
R: We have deleted "the", and please refer to Line 48 in the revised manuscript.

l. 61: spectrometry → spectrometer
R: We have changed the "spectrometry" to "spectrometer", and please refer to Line 51 in the revised manuscript.

l. 63: is not easy → difficult
R: Following the suggestion by Reviewer #2, this sentence have been rephrased to "With the advantages of high precision and stability, GC-MS can distinguish most VOCs qualitatively and quantitatively, however, is difficult in maintaining and operating due to the complex requirements in power, temperature control, and special carrier gas". Please refer to Line 51-53 in the revised manuscript.

l. 91: direction → possibility
R: We have changed the "direction" to "possibility", and please refer to Line 79 in the revised manuscript.

l. 103: space → separation or distance
R: The "space" has been replaced with "distance". Please refer to Line 88 in the revised manuscript.

l. 104: source, → source.
R: The comma has been corrected to the dot. Please refer to Line 90 in the revised manuscript.

l. 109: record spectrum → record the spectrum
R: We have added "the" before "spectrum". Please refer to Line 94 in the revised manuscript.

l. 115: in atmosphere → in the atmosphere
R: We have added "the" before "atmosphere". Please refer to Line 99 in the revised manuscript.

l. 119: fitting differential → fitting the differential
R: We have added "the" between "fitting" and "differential". Please refer to Line 103 in the revised manuscript.

l. 158: Fig. 3 is the linear fitting of the calibration→ Fig. 3 shows the linear fit of the calibration
R: Combined with the suggestion from Reviewer #2, "Fig. 3 is the linear fitting of the calibration" has been changed to "Figure 4 shows the linear fit of the calibration". Please refer to Line 144 in the revised manuscript.

l. 162: For the future → For further
R: We have removed "the", please refer to Line 147 in the revised manuscript.

l. 171 & l. 182: of DOAS → of the DOAS
R: We have added "the" before "DOAS", and please refer to Line 154 and Line 161 in the revised manuscript.

l. 185: with DOAS → with the DOAS
R: We have added "the" before "DOAS", and please refer to Line 167 in the revised manuscript.

l. 189: is lightpath → is the lightpath
R: We have added "the" before "lightpath", and please refer to Line 168 in the revised manuscript.

l. 196: 0.217ppb, → 0.217ppb respectively,
R: "respectively" has been added. Please refer to Line 176 in the revised manuscript.

l. 197: that on-line → that of the on-line
R: We have added "of the" between "that" and "on-line". Please refer to Line 177 in the revised manuscript.

l. 252: isoprene. → isoprene absorption spectrum.
R: We have added "absorption spectrum" after "isoprene". Please refer to Line 226 in the revised manuscript.

**References**

Stutz, J. and Platt U., Numerical analysis and estimation of the statistical error of differential optical absorption spectroscopy measurements with least-squares methods, Applied Optics, 35, 6041-6053, 1996.

Hui, L., Liu, X., Tan, Q., Feng, M., An, J., Qu, Y., Zhang, Y., Cheng, N.: VOC characteristics, sources and contributions to SOA formation during haze events in Wuhan, Central China, Sci of Total Env., 650, 2624-2639, https://doi.org/10.1016/j.scitotenv.2018.10.029, 2019.

EPA, Technical Assistance Document for Sampling and Analysis of Ozone Precursors for the Photochemical Assessment Monitoring Stations Program, U.S. Environmental Protection Agency, EPA-454/B-19-004 (April, 2009).

---

## Author Comment (AC2) · 19 Jan 2021

**Response to reviewers' comments #2**

We thank the reviewers for the constructive comments and suggestions, which are very positive to improve scientific content of the manuscript. We have revised the manuscript appropriately and addressed all the reviewers' comments point-by-point for consideration as below. The remarks from the reviewers are shown in black, and our responses are shown in blue color. All the page and line numbers mentioned following are refer to the revised manuscript without change tracked.

Anonymous Referee #2

The manuscript entitled "Study on the measurement of isoprene by Differential Optical Absorption Spectroscopy" by Song et al. reports the application of DOAS technique on isoprene measurement. This study details the setup, laboratory experiments, and field applications of the DOAS. Intercomparisons of isoprene concentrations measured by the DOAS and a commercial GC-MS shows a good consistency. The content and novelty of the manuscript align well with the requirements of AMT. However, English could be improved throughout the paper as it is not easy to follow. Overall, I recommend the manuscript for publication if the authors can address the following comments.

General Comments:

1. In the calibration experiments described in Sect. 2.2, the actual concentrations (CM) must be measured more than once with parallel experiments for different cell lengths, and measuring error should be added in Table 1 and Figure 3. The equation (1) should be recalculated accordingly. In addition, it seems that the difference between CE and CM increased as the increase of cell length in Table 1. Could authors provide an explanation about the phenomenon?

R: Thank you for the suggestion. In fact, we have performed the parallel experiments for different cell lengths for several times. The measurements for each concentration point are recorded with more than 10 spectra after the system stabilizing. In the revised manuscript, we have indicated all the measured points in the Figure 4 and the standard deviation in Equation (1), as well as added the errors in the Table 2. Please refer to the revised Table 2, Figure 4, and Equation (1) in the revised manuscript.

The calibration results in Table 1 show that the difference between CE and CM increased as the increase of cell length. Even though the absolute difference between CE and CM increased, the relative deviations are constant all through the different concentration points, which can be inferred from the linear regression Figure 4 and Equation (1). As to the absolute difference, it may be due to the possible bias in the length of cell or the error of the standard gas sample.

2. Some important information is missing in the comparison experiments. Firstly, it is not given how reference spectrum was recorded during field applications. As reference spectrum plays a role in spectral fitting, uncertainty caused by reference spectrum should be discussed. Secondly, calibration methods as well as calibration frequency of

the on-line VOCs (TH-300B) are not provided. Compared with the DOAS measurements, the on-line VOCs measurements seems to have a 0.1 ppb offset during the period from 07/21 to 07/24. Could the offset be caused by the calibration of on-line VOCs? Thirdly, providing wind parameters (measured by weather station) and benzene and toluene concentrations (measured by on-line VOCs) when the comparison is inconsistent will be more persuasive, as authors speculated that wind directions and benzene and toluene concentrations would influence the comparison consistency.

R: Thank you for the suggestion. During the field measurement, the measured spectrum collected at 00:00 LT on July 1, 2018 was used as the reference spectrum. The uncertainty caused by reference spectrum have been discussed in Line 104-107 and Line 246-248 of the revised manuscript.

In order to ensure the authenticity and accuracy of the observed data, the working status and response of the TH-300B monitoring system were inspected every day. Daily calibrations were performed automatically at 00:00 to 01:00 LT. In addition, the external standard method for the FID and the internal standard method for the MS were adopted. Please refer to Line 192-197 of the revised manuscript. Regarding to the offset from 21 July, we think it could be explained by two aspects: firstly, the daily calibration operated at midnight could make the on-line VOCs observed value close to the zero point, which may deviate from the real abundance; secondly, the differences of isoprene concertation were existed between the different air masses observed by these two instruments.

Figure R1 shows the discrepancies of measured isoprene by these two instruments as a function of wind direction. It can be seen that the SE and SSE are the prevailing wind direction during the field measurement. Meanwhile, the large difference (defined as abs(DOAS-TH-300B)/DOAS *100%) also tends to appear und SE and SSE wind direction, of which results of discrepancy exceeding 60% were accounted for 54% and 49%, respectively. This suggests that the differences of isoprene observed by these two instruments were impacted by the wind direction.

[Figure]

***Figure R1. Differences of measured isoprene by these two instruments as a function***

*of wind direction.*

Figure R2 shows the dependence of isoprene difference (same as mentioned above) on the benzene and toluene concentration. It can be found that the difference increased as the increases of toluene, however, not obviously benzene. Because the hourly data of benzene and toluene are only representative for the air sampled during dozens of minutes within this hour, it cannot comprehensively reflect their impacts on isoprene.

[Figure]

**Figure R2.** *Dependence of isoprene difference on the benzene and toluene concentration*

However, considering there lots of influencing factors can impact the observed isoprene by DOAS and VOCs analyzer, we are not able to quantify the relationship between observed differences of isoprene and these parameters. Therefore, we have decided to keep these analyses in the responses and provided these possible causes in the manuscript.

3. As authors introduced in Sect. 1, PTR-MS and CIMS can also be used to measure isoprene concentrations. The manuscript would benefit from a critical comparison of the best available performance of these four methods (i.e., DOAS, GC-MS, PTR-MS, and CIMS) together given in a table. Characteristics in the comparison could be time resolution, accuracy, precision, appropriate platforms, etc. Such a comparison would be useful to the readership and meaningful to the community.
R: Thanks for the comments. We have followed the suggestion to summarize a critical comparison of these four methods, as listed in Table R1. Please also refer to Table 1 in the revised manuscript.

*Table R1. Comparison of different on-line methods for isoprene measurement.*

|  | DOAS (this study) | GC-MS (Gong et al., 2018) | PTR-MS (Eerdekens et al., 2009) | CIMS (Leibrock et al., 2003) |
|---|---|---|---|---|
| Time resolution | 1 min | 30-60 min | 0.5-2 min | 1.65 s |

| Accuracy (Correlation with GC-MS/GC) | R=0.85 | R>0.99 (with offline) | 0.95 | R=0.78 |
|---|---|---|---|---|
| Detection Limit | 10 ppt | 4 ppt | 100 ppt | <30 ppt |
| Platform | Stationary / conditional mobile | Stationary / mobile | Stationary / mobile | Stationary / mobile |
| Advantages | No sampling Easy operation Simple instrument | High precision Accurate quantification | Fast responses High precision | High time resolution Good sensitivity |
| Disadvantages | Impacts by weather conditions Impacts of interferences | Time consuming Calibration needed Difficult operating and maintaining | Molecule or fragment ion of the same mass cannot be differentiated | Interference of unidentified components Expensive equipment |

Specific Comments:

Line 15: "202.71-227.72nm" → "202.71-227.72 nm". Blank space should be inserted between number and unit. Such irregular expressions were used frequently elsewhere in the manuscript and should be revised.

R: Thanks for the comments. We have corrected for the irregular expressions all through the manuscript and the blank space has been inserted correspondingly in Line 14 and elsewhere in the revised manuscript.

Line 26: The "B" in BVOCs is usually the abbreviation of "Biogenic" instead of "Biological".

R: Thank you for the suggestion. The "Biological" has been replaced with "Biogenic". Please refer to Line 23 in the revised manuscript.

Line 42: "In the daytime, the oxidation by OH is the main chemical process of isoprene." The sentence should specify "whose oxidation" and "what kind of chemical process" to avoid ambiguous meaning.

R: Thank you for the suggestion. We have specified this sentence as "In the daytime, the chemical process oxidized by OH is the main sink of isoprene". Please refer to Line 35-36 in the revised manuscript.

Line 44-47: These contents are given here without references.

R: Thank you for the suggestion. Relevant literatures have been cited there, e.g. Chen et al., 2020; Lu et al., 2018; Zhu et al., 2020. Please refer to Line 39 and References in the revised manuscript.

Line 52: "BVOCs also has [: : :]" → "BVOCs has [: : :]"

R: We have deleted the unnecessary "also". Please refer to Line 43 in the revised manuscript.

Line 59: "GC-MS is using the high separation ability of gas chromatography to separate the [: : :]". Simple Present Tense should be used here.

R: Thanks for the comments. Simple present tense has been used to rephrase the sentence as "GC-MS utilizes the high separation ability of gas chromatography to separate the components of environmental samples, and then measures the different compounds with the mass spectrometry". Please refer to Line 50-51 in the revised manuscript.

Line 61-64: "Although GC-MS [: : :] But the complex [: : :]" These sentences should be rephased.

R: Thank you for the suggestion. These sentences have been rephrased to "With the advantages of high precision and stability, GC-MS can distinguish most VOCs qualitatively and quantitatively, however, is difficult in maintaining and operating due to the complex requirements in power, temperature control, and special carrier gas". Please refer to Line 51-53 in the revised manuscript.

Line 63ff: A comma should go before the conjunction "and" in a list of three or more items. "[: : :] in power, temperature control and special carrier gas [: : :]" → "power, temperature control, and special carrier gas". "[: : :] requires sampling, preservation and pre-treatment [: : :]" → "requires sampling, preservation, and pre-treatment".

R: Thanks for the comments. We have added the comma in the corresponding places. Please refer to Line 53 and 54 in the revised manuscript.

Line 67, 77, and 78: The meaning of the abbreviations (i.e., PTR-MS, GC, and CIMS) has already been given in the previous paragraph and so it need not be defined again here.

R: Thank you for the suggestion. The abbreviations were used there. Please refer to Line 57, 63, and 66 in the revised manuscript.

Line 69-71: The sentences should be rephased.

R: Thank you for the suggestion. We have re-written as "The fixed length of the drift tube provides a fixed reaction time for the ions as they move along the drift tube. The sample air is continuously pumped through the drift tube and the VOCs in the sample react with $H_3O^+$ to be ionized, and then enter the mass spectrometer to be detected". Please refer to Line 58-60 in the revised manuscript.

Line 104: "[: : :] as light source, the aperture [: : :]" → "as light source. The aperture"

R: Thank you for the suggestion. We have corrected it and please refer to Line 90 in the revised manuscript.

Line 106: "while, the aperture of the" → "while the aperture of the"
R: Thanks for the comments. Combined with the suggestions by Reviewer #1, we have rephrased these sentences. Please refer to Line 90-92 in the revised manuscript.

Line 109: "1024-pixel photodiode array as detector was used to record spectrum" → "1024-pixel photodiode array was used as detector to record spectrum"
R: Thanks for the comments. We have corrected this sentence. Please refer to Line 93-94 in the revised manuscript.

Line 158: "Fig. 3" → "Figure 3"
R: Thanks for the comments. We have corrected it and please refer to Line 144 in the revised manuscript.

Line 215: "[: : :] so higher measurement points will catch higher concentration of isoprene. [: : :]" Reference or detailed explanations should be given here.
R: As pointed by Reviewer #1, we found that this statement is wrong and remove it from the revised manuscript.

Line 217: "[: : :] will be more or less lost during the sampling process." Sampling loss of on-line VOCs is an important parameter which should be quantized here by performing experiments or referring to a similar research.
R: The TH-300B on-line VOC instrument uses detection technology that includes ultralow-temperature preconcentration combined with gas chromatography and mass spectrometry (GC/MS). For a complete measuring cycle, there five steps include sample collection, freeze-trapping, thermal desorption, GC-FID/MS analysis, heating and anti-blowing purification. It takes about 1h for one complete detection cycle. It's extremely difficult to evaluate the sampling loss individually from the complete detection cycle, especially for experiments. As a commercial scientific instrument, the relative error of the targets was quantized not exceeding 30% among all the detected VOCs species for the whole procedure, which is determined by the difference between the measured and guaranteed values of standard samples. As to isoprene, the $R^2$ reached 0.999 (Hui et al., 2019). It suggests that the detection accuracy is very high, and the total error could be contributed considerably by sampling process. We have addressed the related discussion in Line 191-192 in the revised manuscript.

Line 241-243: The authors should provide an explanation or references on the method that they used to calculate detection limit.
R: Here we have cited the National Environmental Protection Standard HJ654-2013, in which the detection limit of ambient air quality continuous automated monitoring system using open light path method, i.e. DOAS in this study, can be defined as the two times of zero noise. Please refer to Line 215 to 217 in the revised manuscript.

Line 247-249: As the stability of light source and spectrometer will influence the fitting

residual and instrumental performance, sensitivity experiments of temperature (or other relative parameters) for light source and spectrometer should be conducted.

R: Thank you for the suggestion. Considering the influences of light source and spectrometer stability on the fitting residual and instrumental performance, we have used the thermostatic apparatus to keep the operating temperature of spectrometer stable. And the air conditioner serves as thermostat model to stable the ambient temperature for the whole system of DOAS instrument. The record of thermometer show that the ambient temperature varied within ±1 ℃. We have indicated related description in Line 222-224 in the revised manuscript.

Line 293: "CIMS methods, The" → "CIMS methods, the"
R: Thanks for the comments. It has been corrected and please refer to Line 263 in the revised manuscript.

Line 304-306: The sentences should be rephased.
R: Thank you for the suggestion. We have rephrased the sentence as "Considering the differences in measurement principle and sampled air between them, the comparison results show a good agreement between these two instruments". Please refer to Line 271-273 in the revised manuscript.

Line 308: "the paper proposes [: : :]" → "this study proposes [: : :]"
R: Thanks for the comments. "the paper" has been changed to "this study". Please refer to the Line 275 in the revised manuscript.

**References**

Gong, D., Wang, H., Zhang, S., Wang, Y., Liu, S. C., Guo, H., Shao, M., He, C., Chen, D., He, L., Zhou, L., Morawska, L., Zhang, Y., and Wang, B.: Low-level summertime isoprene observed at a forested mountaintop site in southern China: implications for strong regional atmospheric oxidative capacity, Atmos. Chem. Phys., 18, 14417–14432, https://doi.org/10.5194/acp-18-14417-2018, 2018.

Eerdekens, G., Ganzeveld, L., Vilà-Guerau de Arellano, J., Klüpfel, T., Sinha, V., Yassaa, N., Williams, J., Harder, H., Kubistin, D., Martinez, M., and Lelieveld, J.: Flux estimates of isoprene, methanol and acetone from airborne PTR-MS measurements over the tropical rainforest during the GABRIEL 2005 campaign, Atmos. Chem. Phys., 9, 4207–4227, https://doi.org/10.5194/acp-9-4207-2009, 2009.

Leibrock, E., Huey, L. G., Goldan, P. D., Kuster, W. C., Williams, E., and Fehsenfeld, F. C.: Ground-based intercomparison of two isoprene measurement techniques, Atmos. Chem. Phys., 3, 67–72, https://doi.org/10.5194/acp-3-67-2003, 2003.

Hui, L., Liu, X., Tan, Q., Feng, M., An, J., Qu, Y., Zhang, Y., Cheng, N.: VOC characteristics, sources and contributions to SOA formation during haze events in Wuhan, Central China, Sci of Total Env., 650, 2624-2639, https://doi.org/10.1016/j.scitotenv.2018.10.029, 2019.

Chen, T., Xue, L., Zheng, P., Zhang, Y., Liu, Y., Sun, J., Han, G., Li, H., Zhang, X., Li, Y., Li, H., Dong, C., Xu, F., Zhang, Q., and Wang, W.: Volatile organic compounds and ozone air pollution in an oil

production region in northern China, Atmos. Chem. Phys., 20, 7069–7086, https://doi.org/10.5194/acp-20-7069-2020, 2020.

Lu, K., Guo, S., Tan, Z., Wang, H., Shang, D., Liu, Y., Li, X., Wu, Z., Hu, M., and Zhang, Y.: Exploring atmospheric free-radical chemistry in China: the self-cleansing capacity and the formation of secondary air pollution, Natl. Sci. Rev., 6, 579–594, https://doi.org/10.1093/nsr/nwy073, 2018.

Zhu, J., Wang, S., Wang, H., Jing, S., Lou, S., Saiz-Lopez, A., and Zhou, B.: Observationally constrained modeling of atmospheric oxidation capacity and photochemical reactivity in Shanghai, China, Atmos. Chem. Phys., 20, 1217–1232, https://doi.org/10.5194/acp-20-1217-2020, 2020.